# Intestinal Epithelial Creatine Transporter SLC6A8 Dysregulation in Inflammation and in Response to Adherent Invasive *E. coli* Infection

**DOI:** 10.3390/ijms25126537

**Published:** 2024-06-13

**Authors:** Harshal Sawant, Rajesh Selvaraj, Prasath Manogaran, Alip Borthakur

**Affiliations:** Department of Clinical and Translational Sciences, Joan C. Edwards School of Medicine, Marshall University, Huntington, WV 25701, USA; sawantha@marshall.edu (H.S.); selvarajr@marshall.edu (R.S.); prasathbiomed@gmail.com (P.M.)

**Keywords:** Crohn’s disease, intestinal inflammation, creatine transporter, intestinal organoids

## Abstract

Creatine transporter (CrT1) mediates cellular uptake of creatine (Cr), a nutrient pivotal in maintaining energy homeostasis in various tissues including intestinal epithelial cells (IECs). The impact of CrT1 deficiency on the pathogenesis of various psychiatric and neurological disorders has been extensively investigated. However, there are no studies on its regulation in IECs in health and disease. Current studies have determined differential expression of CrT1 along the length of the mammalian intestine and its dysregulation in inflammatory bowel disease (IBD)-associated inflammation and Adherent Invasive *E. coli* (AIEC) infection. CrT1 mRNA and protein levels in normal intestines and their alterations in inflammation and following AIEC infection were determined in vitro in model IECs (Caco-2/IEC-6) and in vivo in SAMP1/YitFc mice, a model of spontaneous ileitis resembling human IBD. CrT1 is differentially expressed in different regions of mammalian intestines with its highest expression in jejunum. In vitro, CrT1 function (Na^+^-dependent ^14^C-Cr uptake), expression and promoter activity significantly decreased following TNFα/IL1β treatments and AIEC infection. SAMP1 mice and ileal organoids generated from SAMP1 mice also showed decreased CrT1 mRNA and protein compared to AKR controls. Our studies suggest that Cr deficiency in IECs secondary to CrT1 dysregulation could be a key factor contributing to IBD pathogenesis.

## 1. Introduction

Inflammatory bowel diseases (IBD), including ulcerative colitis (UC) and Crohn’s disease (CD), are chronic relapsing–remitting inflammatory disorders of the gastrointestinal tract [1,2]. IBD occurs in genetically susceptible individuals after an exaggerated immune response upon exposure to environmental triggers [3,4]. Altered dietary habits and other environmental factors that cause alteration in gut microbiota critically affect IBD progression and recurrence [5,6,7,8]. Further, for its relapsing–remitting episodic nature, IBD management requires disease-specific long-term preventive care with an effective diet and lifestyle plan.

Following the “starved gut hypothesis” [9,10], IBD has been considered a disease of diminished mucosal nutrition and energy deficiency (starved gut) that strongly coincides with the degree of inflammation. Subsequently, it has been appreciated that IBD-associated shifts in the microbiota, termed dysbiosis, cause inordinate demands for energy acquisition within the mucosa, particularly during active inflammation [11,12]. Impaired nutrients and electrolyte absorption in IBD secondary to dysregulation of the respective nutrient/electrolyte transporters have been reported in previous studies by us and others [13,14,15,16,17,18,19]. However, the role of energy deficiency in IBD pathogenesis is not well understood. Recent studies have implicated that storage pools of energy may also be deficient in IBD [11]. In this regard, creatine (Cr) has emerged as a safe nutritional supplement that strengthens cellular energetics for its essential role in energy metabolism in different tissues and has conferred beneficial effects in multiple pathologies, such as myopathies or neurodegenerative diseases [20]. These energy reserves are provided largely by the creatine–phosphocreatine shuttle, and recent investigations have defined the important role of Cr in intestinal inflammation [21]. Approximately 50% of Cr is obtained from the diet, supporting the rationale for Cr supplementation as a potential adjuvant therapy for IBD [21]. In IECs, Cr has also been shown to play a pivotal role in strengthening epithelial barrier function, a vital energy-requiring process compromised in IBD [22,23]. However, the mechanisms of Cr’s effects in augmenting cellular energy and barrier function in IECs and the potential ameliorative role of Cr in IBD are not well investigated. Further, the pleiotropic beneficial effects of Cr in IBD depend on how efficiently IECs in the inflamed mucosa of IBD patients absorb dietary or supplemental Cr. The creatine transporter (CrT1, SLC6A8), expressed on the apical membrane of IECs, mediates Na^+^-dependent uptake of dietary Cr with high affinity for Cr [24]. A recent study reported a significant decrease in CrT1 expression in the intestinal mucosa of IBD patients [23]. However, the underlying mechanisms are not known. Since optimal Cr absorption into IECs by CrT1 is critical for energy homeostasis and barrier function, understanding the mechanisms of dysregulation of CrT1 in IBD is critical in evaluating the potential of CrT1 as a novel therapeutic target for IBD.

Adherent-invasive *Escherichia coli* (AIEC), an opportunistic pathogen that commonly colonizes the intestinal mucosa of Crohn’s disease (CD) patients has recently been implicated as a causative or contributory factor in CD pathology [25]. AIEC is enriched in the terminal ileum of CD patients and has a strong association with disease inflammatory activity and postoperative recurrence [26,27]. Colonization of AIEC has also been shown in experimental animal models causing both acute and chronic intestinal inflammation resembling human IBD pathology [28,29]. There are several studies investigating epithelial barrier disruption following AIEC infection [30]. However, despite mucosal nutrient and energy deficiency being the hallmark of IBD, how AIEC infection could affect epithelial nutrient transport, including CrT1-mediated creatine transport, is not known.

Here, we report for the first time the differential expression of CrT1 along the length of mouse, rat, and human intestines and demonstrate significant downregulation of CrT1 expression and function in IBD-associated inflammation and in response to AIEC infection via distinct mechanisms.

## 2. Results

### 2.1. mRNA and Protein Levels of CrT1 along the Length of Rat, Mouse, and Human Intestines

The scrapped mucosa from different regions of 18-week-old Sprague Dawley rats and 10-week-old C57B/6 mice and jejunal and colonic mucosal biopsies of healthy human subjects were used for total RNA extraction and lysate preparation. The mRNA and protein levels of CrT1 in total RNA and mucosal whole-cell lysates, respectively, were determined by real-time quantitative RT PCR and Western blot, utilizing gene-specific primers, or appropriate antibodies. As shown in Figure 1, CrT1 mRNA was differentially expressed in different regions of mouse (A) and rat (B) intestines, showing the highest levels in the jejunum.

Differential expression of the CrT1 protein in different regions of mouse (A), rat (B), and human (C) intestines is shown in Figure 2. Consistent with the mRNA results, the highest CrT1 protein levels were found in the jejunum in all species. The observed subtle difference in the band intensities of the CrT1 protein across the different species was presumably due to variations in the amount of protein (µg) loaded per well for mouse, rat, and human samples, which can be found in the figure legend.

### 2.2. CrT1 Is Predominantly Localized on the Luminal Surface of Enterocytes

For optimal absorption of dietary nutrients from the gut lumen, the nutrient transporters need to be expressed and localized to the lumen-facing brush border membranes (BBM) of villus cells. Since the CrT1 protein level was the highest in the jejunum, we used immunofluorescence studies in rat jejunal mucosal cryosections to determine the localization of CrT1 in the villus cells. As shown in Figure 3, CrT1 is expressed uniformly along the length of the villi and localized on the luminal (brush border) membrane of the villus cells. To further confirm its BBM localization, we measured the colocalization of CrT1 with villin, a BBM marker protein. The merged image clearly shows the colocalization of CrT1 with villin in the BBM of villus cells.

### 2.3. CrT1 Function and Expression Are Decreased in Inflammation

A recent study reported reduced expressions of CrT1 in the colonic mucosa of IBD patients [23]. Since the absorption of dietary nutrients, including creatine (Cr), occurs in the small intestine (jejunum and ileum) and since we found the highest expression of CrT1 in the jejunum, this study focused on determining the effects of inflammation on the function and expression of small intestinal CrT1. Therefore, to study CrT1 dysregulation in inflammation, we used the rat small intestinal cell line IEC-6 and human colonic cell line Caco-2, which when grown as fully differentiated post-confluent monolayers, morphologically and functionally mimic absorptive enterocytes [31]. Caco-2/IEC-6-confluent monolayers grown on transwell inserts were treated from basolateral surfaces with TNFα or IL1β (10 ng/mL), proinflammatory cytokines known to be elevated in IBD, for 24 h. CrT1 function (Na^+^-dependent ^14^C-creatine uptake) and expression (mRNA and protein) were measured as described in Methods. CrT1 function (Na^+^-dependent ^14^C-creatine uptake) was significantly decreased by both cytokines in both cell lines (Figure 4).

Similarly, mRNA (A) and protein (B) levels of CrT1 were significantly decreased in both cell lines in response to TNFα or IL1β treatments (Figure 5).

### 2.4. TNFα and IL1β Treatments Decrease CrT1 Promoter Activity in Caco-2 Cells

Since both mRNA and protein levels of CrT1 were decreased following TNFα/IL1β treatments, we sought to determine if the cytokine-mediated decrease in CrT1 expression is at the level of transcription. The 1029 bp human CrT1 promoter obtained from Active Motif was transfected into Caco-2 cells, and promoter activity was measured as described in Methods. As shown in Figure 6, CrT1 promoter activity significantly decreased following TNFα or IL1β treatments. These results suggest that TNFα and IL1β treatments reduce CrT1 expression via transcriptional mechanisms.

### 2.5. CrT1 Expression Is Decreased In Vivo in Ileitis

Next, we used SAMP1/YitFc mice, a model of spontaneous ileitis closely resembling human Crohn’s disease, to measure small intestinal CrT1 expression compared to AKR/J control mice. Both mRNA (A) and protein (B) levels of CrT1 were decreased in SAMP1 mice compared to the AKR controls (Figure 7).

### 2.6. CrT1 Expression Is Decreased in SAMP1 Small Intestinal Organoids Compared to AKR Organoids

Small intestinal crypt-derived organoids were prepared to be utilized as a physiologically relevant in vitro model of the native intestine to study CrT1 regulation. Like SAMP1 small intestinal mucosa, mRNA (A) and protein (B) levels of CrT1 were significantly decreased in SAMP1 organoids compared to AKR organoids (Figure 8). These results validate the suitability of SAMP1 versus AKR organoids as an appropriate model to study the mechanisms of CrT1 dysregulation in IBD-associated inflammation.

### 2.7. AIEC Infection Decreases CrT1 Expression In Vitro

Adherent-invasive *Escherichia coli* (AIEC), an opportunistic pathogen implicated as a causative or contributory factor in Crohn’s disease (CD), has been described to colonize the gut of 40% of CD patients [32]. AIEC is enriched in the terminal ileum of CD patients and has a strong association with disease inflammatory activity and postoperative recurrence [26,27]. AIEC colonization has also been shown in animal models of IBD, causing both acute and chronic intestinal inflammation and exacerbating pre-existing inflammation [28,29]. Therefore, we have examined if AIEC infection affects CrT1 expression. IEC-6 and Caco-2 monolayers were treated with LF-82, a pathogenic and invasive AIEC strain for 8 h. As shown in Figure 9, CrT1 mRNA levels significantly decreased in both cell lines following LF-82 infection.

### 2.8. AIEC Infection Diminishes CrT1 in SAMP1 Organoid-Derived Monolayers

2D monolayers derived from SAMP1 and AKR 3D organoids were infected with LF-82 for 8 h. CrT1 mRNA was significantly decreased in SAMP1 monolayers following LF-82 infection, with no significant effect on CrT1 mRNA in AKR monolayers (Figure 10). These results suggest that AIEC effects on CrT1 are more pronounced in mucosa with pre-existing inflammation.

## 3. Discussion

Creatine (Cr), a non-protein amino acid, is mainly known for its essential role in energy metabolism in different tissues [20]. Creatine kinase (CK) catalyzes the reversible conversion of Cr and ATP to phosphocreatine and ADP. Thus, the creatine–phosphocreatine (Cr-pCr) system serves as a shuttle of high-energy phosphates between mitochondrial sites of production and cytosolic sites of utilization and mediates a critical energy distribution mechanism in all cells with high energy demand [33], including intestinal epithelial cells (IECs). Cr has become a major nutritional supplement in sports medicine [34,35] and is intensively investigated as a therapeutic agent in psychiatric disorders and various neurological conditions (mitochondrial encephalopathy, stroke, traumatic neurological injury, and neurodegenerative and muscular disorders) [36,37]. Cellular entry of Cr is mediated by a creatine transporter (CrT1) differentially expressed in various tissues. More than 50% of this essential energy nutrient comes from the diet, e.g., meat and fish. Dietary or supplemental Cr absorbed by the IECs is utilized to maintain energy homeostasis in IECs and various other body tissues.

A specific CrT1, encoded by the X-linked gene *SLC6A8*, and a member of the SLC (solute carrier) 6 family, is present in the lumen-facing apical membranes of IECs which mediates the uptake of dietary Cr with a high affinity (K_m_ for Cr of 30 µM) [24]. SLC6A8 is ubiquitously expressed in various tissues with high energy demand, mediating Cr uptake from systemic circulation. The regulation of SLC6A8 has been extensively investigated in brain, heart, muscle, and kidney tissues [38,39,40]. However, despite its paramount importance in assimilating dietary or supplemental Cr, there are virtually no studies on the mechanisms regulating intestinal epithelial SLC6A8 in health and disease. Similarly, the effects of SLC6A8 deficiency and/or dysregulation in the occurrence of various psychiatric disorders and neurological conditions have been extensively investigated [41,42]. However, how CrT1 deficiency could disrupt intestinal epithelial energy homeostasis, more particularly in the pathogenesis of inflammatory bowel diseases (IBD) manifested by mucosal energy deficiency, has only recently been appreciated [23]. Therefore, in the current study, we have initially determined the relative expression of CrT1 along the length of the mammalian (mouse, rat, and human) intestine and then elucidated the mechanisms of its dysregulation in Crohn’s disease (CD), a major form of IBD primarily affecting the small intestine. We found the highest levels of CrT1 mRNA and proteins in the jejunum followed by the ileum, the segments of the intestine instrumental in the absorption of dietary nutrients. To determine the mechanisms of dysregulation of CrT1 in IBD-associated inflammation, we utilized SAMP1/YitFc mice, an inbred mouse strain that develops ileitis spontaneously, with AKR/J mice, from which SAMP1 mice originated, as the control. SAMP1/YitFc mice possess remarkable similarities to the human CD regarding disease location and histologic features and display a well-defined time course of a pre-disease state and phases of acute and chronic ileitis [43]. Since most of the nutrients including Cr are absorbed in the small intestine and since our initial data have shown the highest CrT1 expression in the jejunum followed by ileum, for in vitro studies we used rat intestinal cell line IEC-6 and human intestinal cell line Caco-2. IEC-6 is a small intestinal cell line, whereas Caco-2, a colon cancer-derived cell line, when grown as fully differentiated post-confluent monolayers, morphologically and functionally mimics absorptive enterocytes [31]. In both cell lines, CrT1 function and expression were determined following treatments with IBD-associated proinflammatory cytokines TNFα and IL1β. Altered CrT1 expression was also measured in small intestinal crypt-derived organoids generated from SAMP1 mice compared to AKR organoids. Our results in these multiple models of inflammation (SAMP1 mice and SAMP1 organoids, IEC-6/Caco-2 cells treated with TNFα/IL1β) has, for the first time, demonstrated a significant decrease in both mRNA and protein levels of CrT1. In vitro, TNFα and IL1β treatments of Caco-2 cells significantly decreased CrT1 promoter activity, suggesting that the inflammatory mediators decrease CrT1 expression via modulating CrT1 gene transcription.

Since Roediger et al. proposed the “starved gut hypothesis”, IBD has been considered a disease of diminished mucosal nutrition and energy deficiency (starved gut) that strongly coincides with the degree of inflammation [9,10]. In particular, the most widely appreciated IBD-associated shifts in gut microbiota, termed dysbiosis, have been implicated in mucosal energy deficiency, presumably as a result of mitochondrial dysfunction. This, in turn, is likely to cause inordinate demands for energy acquisition within the mucosa, particularly during active inflammation [11]. However, how mucosal energy deficiency triggers chronic inflammation in IBD pathogenesis is not well understood. Recent studies have implied that the storage pools of energy (energy reserves), provided largely by the creatine–phosphocreatine shuttle, may also be deficient in IBD. Therefore, our studies demonstrating an extensive downregulation of CrT1 in inflamed mucosa, which could severely impair Cr availability in IECs, highlight a novel facet of IBD pathogenesis involving mucosal energy deficiency (Figure 11). Whether CrT1 downregulation is a cause or consequence of IBD-associated chronic inflammation, however, is not evident from our current study results.

Since adherent-invasive *Escherichia coli* (AIEC), an opportunistic pathogen, has recently been implicated as a causative or contributory factor in CD pathology [25], we sought to determine if AIEC infection could be involved in prolonging the chronic inflammation in IBD via further worsening the mucosal energy deficient state. Our in vitro results in IEC-6/Caco-2 cells showed a significant decrease in CrT1 mRNA and protein levels following AIEC infection for 24 h. We next determined the effects of AIEC infection on CrT1 expression in 2D monolayers prepared from SAMP1 versus AKR 3D organoids. It is important to note that CrT1 mRNA was significantly decreased in SAMP1 organoid monolayers following AIEC LF-82 infection, with no significant effect on CrT1 mRNA in AKR monolayers. These results suggest that AIEC effects on CrT1 are more pronounced in mucosa with pre-existing inflammation. Thus, our studies have for the first time demonstrated AIEC infection effects on an intestinal epithelial nutrient transporter implicated in IBD pathogenesis.

Based on theoretical considerations, experimental data in vitro and from animal models, and a single-case pilot study with one colitis patient [44], a consolidated hypothesis has recently been put forward that oral supplementation of Cr, a pleiotropic cellular energy precursor, could be effective in inducing favorable response and/or remission in IBD [21]. However, our current study and a recent study by others [23] showing defective Cr transport into IECs in IBD emphasize the need for extensive mechanical studies in experimental models supported by clinical cohort studies in IBD (both CD and UC) patients to unravel the chicken-and-egg story: whether CrT1 deficiency is a cause or consequence of IBD-associated inflammation. This will help design combined therapies for the prevention and/or remission of IBD.

## 4. Materials and Methods

### 4.1. Animal Studies and Human Intestinal Biopsy Samples

Our animal studies were approved by the Institutional Animal Care and Use Committee (IACUC) of Marshall University. Male C57B/6 mice (Jackson Laboratories, Farmington, CT, USA), Sprague Dawley rats (Charles River Laboratories, Wilmington, MA, USA) (under IACUC protocol #756), and SAMP1/YitFc (SAMP1) and age-matched control AKR/J (AKR) mice (Jackson Laboratories) (under IACUC protocol #743) were maintained in a 12 h light/dark cycle with free access to food and water in the Animal Care Facility at Byrd Biotechnology Science Center, Marshall University. After a week of acclimatization, the animals were included in the study at 10 weeks of age for mice and 18 weeks of age for rats to measure the expression and localization of CrT1 in various regions of the intestine. Scrapped mucosa from the jejunum, ileum, proximal, and distal colon were snap-frozen and stored at −80 °C for further analyses of mRNA and protein levels of CrT1. Altered expression of CrT1 in IBD was determined in SAMP1/YitFc mice, a model of spontaneous ileitis resembling human IBD (27) with normal AKR/J mice as controls.

Jejunal and colonic mucosal biopsies of healthy human subjects were obtained from the Translational Core of the Department of Clinical and Translational Science, Marshall University, under an approved IRB (IRBNet ID #964144-18).

### 4.2. Generation of Crypt-Derived Organoids and Organoid-Derived Monolayers

Small intestinal crypt-derived organoids were prepared from AKR and SAMP1 mice according to the protocol of Sato et al. [45] with lab-specific modifications previously described by us [46]. Organoid-derived 2D monolayers were generated from 3D organoids using a protocol adapted from Ettayebi et al. [47] with lab-specific, minor modifications.

### 4.3. Cell Culture and Treatment

Model human intestinal Caco-2 cells and rat intestinal IEC-6 cells, both obtained from American Type Culture Collection (ATCC, Manassas, VA, USA), were cultured following the conditions described previously by us [17,48]. Briefly, Caco-2 cells were grown in MEM supplemented with 50 U/mL penicillin, 50 µg/mL streptomycin, 50 µg/mL gentamicin, and 20% fetal bovine serum (FBS) at 37 °C in a 5% CO_2_-95% air environment. IEC-6 cells were grown in DMEM with 4.5 g/l glucose, 50 U/mL penicillin, 50 µg/mL streptomycin, 50 µg/mL gentamicin, and 10% FBS. For treatments with proinflammatory agents (TNF-α and IL-1β), receptors of which are expressed on the basolateral surface of IECs, the cell monolayers were grown on 12-well transwell inserts. For AIEC infection, cell monolayers were grown on 6-well plastic supports.

### 4.4. Bacterial Cultures and Treatments of Cell Monolayers and Organoid-Derived Monolayers

The pathogenic AIEC strain LF-82 (obtained from Dr. Kenneth William Simpson, Cornell University, Ithaca, NY, USA) was grown overnight by inoculating single colonies in a BHI (brain–heart infusion) medium (MP Biomedicals, Solon, OH, USA). Overnight cultures grown exponentially in a BHI medium to attain an optical density (absorbance) at 600 nm of 0.2 were diluted 1:10 with cell culture media (DMEM/F12) and applied to the apical surface of cell or organoid monolayers for time periods as indicated in the figure legends.

### 4.5. Measurement of CrT1 Function

CrT1 function was measured as Na^+^-dependent uptake of [^14^C] creatine as described previously [38]. Briefly, polarized IECs grown on transwell filters were serum-starved for 30 min at 37 °C in Hanks’ buffer and then washed twice with Krebs-Ringer–HEPES (KRH) buffer. Uptake assays were performed by apical application of KRH buffer containing 10 μM [^14^C] creatine (10 mCi/mmol; American Radiolabeled Chemicals Inc) for 30 min at 37 °C (*n* = 3 for each condition). The competitive CrT1 substrate β-guanidinopropionic acid (GPA, 1 mM) was added in some fluxes to eliminate background non-CrT1-mediated uptake. The flux was terminated by aspirating the uptake medium followed by washing twice with 0.5 mL of ice-cold KRH buffer in which 120 mM LiCl was subsequently substituted for NaCl. Cells were lysed with 80 µL of lysis buffer. An amount of 50 µL of lysate was counted in a liquid scintillation counter. Results normalized by protein concentrations (determined by the Bradford technique) were expressed as nmol [^14^C] creatine/mg protein/30 min.

### 4.6. Real-Time Quantitative Polymerase Chain Reaction

RNA was extracted from mouse and rat intestinal mucosal samples, mouse organoid monolayers, and Caco-2 and IEC-6 cells using RNeasy kits (Qiagen, Germantown, MD, USA). RNA was reverse-transcribed and amplified using a Brilliant SYBR Green QRT-PCR Master Mix kit (Agilent Technology, Santa Clara, CA, USA). Human, mouse, and rat CrT1 was amplified with gene-specific primers custom-designed by Thermo Fisher Scientific (Hillsboro, OR, USA) with GAPDH or β2-microglobulin (for organoids) amplified as the internal controls. The sequences of the primers used in this study are given in Table 1.

### 4.7. Western Blotting

Tissue lysates from the scraped intestinal mucosa or cell lysates were prepared using RIPA buffer (Thermo Fisher Scientific). The lysates were run on 10% SDS-PAGE and then transferred onto PVDF membranes. Immunoblotting was carried out with anti-CrT1 antibody (Proteintech, Rosemont, IL, USA, Catalog #20299-1-AP, dilution 1:1000) and with β-actin antibody (Santa Cruz Biotechnology, Santa Cruz, CA, USA, Catalog #sc47778, dilution 1:2000) as the loading control. Bands were visualized with enhanced chemiluminescence detection reagents. Relative band intensities were measured utilizing Image Lab software, version 6.1 (BioRad, Hercules, CA, USA).

### 4.8. Promoter Activity

A 1029 bp fragment of the human CrT1 promoter region cloned into pLightSwitch_Prom (Active Motif, Carlsbad, CA, USA) was transfected into Caco-2 cells utilizing Lipofectamine 3000 transfection reagent (Thermo Fisher) according to the manufacturer’s protocol. Promoter activity was determined utilizing LightSwitch luciferase assay kit (Active Motif), according to the manufacturer’s protocol.

### 4.9. Immunofluorescence Studies in Mouse and Rat Mucosa

Immunofluorescence studies in mucosal sections were performed as described previously by us [48]. Briefly, sections of small intestinal tissues from mice and rat groups were snap-frozen in optimal-cutting-temperature (OCT) embedding medium. For immunostaining, 5 μm frozen sections were fixed with 1% paraformaldehyde in PBS for 10 min at room temperature. Fixed sections were washed in PBS, permeabilized with 5% Nonidet P-40 for 5 min, and blocked with 5% normal goat serum (NGS) for 30 min. The tissues were incubated with CrT1 antibody (1:100) in PBS with 1% NGS for 90 min at room temperature. After being washed, the sections were incubated with Alexa Fluor 594-conjugated goat anti-rabbit IgG and Alexa Fluor 488-conjugated phalloidin (5 U/mL; Invitrogen, Waltham, MA, USA) for 60 min. The sections were then washed and mounted under coverslips using Slowfade Gold antifade with DAPI reagent (Invitrogen). The sections were imaged using a Carl Zeiss LSM 510 laser-scanning confocal microscope equipped with a ×20 water immersion objective.

### 4.10. Data Analysis and Statistics

Excel version 16.85 (Microsoft, Redmond, USA), Image Lab 5.2.1 (BioRad Laboratories, Hercules, CA, USA), and Prism 8.0 (GraphPad Software, San Diego, CA, USA) software were used for data acquisition, analysis, and presentation. Tests used for the data analysis in Prism were unpaired *t*-tests and one-way and multifactorial Analysis of Variance (ANOVA) depending on the data sets, as appropriate. Error bars indicated ± Standard Error of Mean (SEM). Probability values of *p* < 0.05 were considered statistically significant. The Shapiro–Wilk test in Prism was used to confirm a normal distribution of the data sets.

## 5. Conclusions

Creatine transport into various tissues mediated by the creatine transporter (CrT1) is pivotal in maintaining cellular energy homeostasis, thereby ensuring proper functioning of vital energy-requiring cellular processes. Our studies have, for the first time, demonstrated in multiple experimental models that CrT1, differentially expressed along the length of the mammalian intestine, is downregulated in inflamed mucosa in IBD via distinct mechanisms. CrT1 could be further downregulated following infection with AIEC, an opportunistic pathogen increasingly being appreciated to be closely linked to Crohn’s disease pathology, culminating in a state of mucosal energy deficiency. Therefore, our studies highlight CrT1 as a novel therapeutic target for IBD and support the rationale for an urgent need for extensive cohort studies with IBD patients to establish creatine supplementation as an effective adjuvant therapy treating IBD.

## Figures and Tables

**Figure 1 ijms-25-06537-f001:**
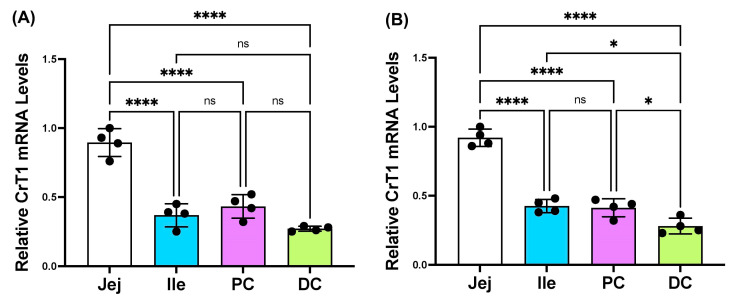
mRNA levels of CrT1 along the length of mouse (**A**) and rat (**B**) intestinal mucosa. Total RNA isolated from scrapped mucosa was amplified with species (mouse or rat)-specific gene (*CrT1*) primers for real-time PCR quantification. Data represent the relative expression of CrT1 normalized to respective GAPDH mRNA (internal control) in different segments of the intestine (Jej: jejunum; Ile: ileum; PC: proximal colon; DC: distal colon). Data represent mean ± SEM. * *p* < 0.05, **** *p* < 0.0001 and ns = not significant between groups as indicated.

**Figure 2 ijms-25-06537-f002:**
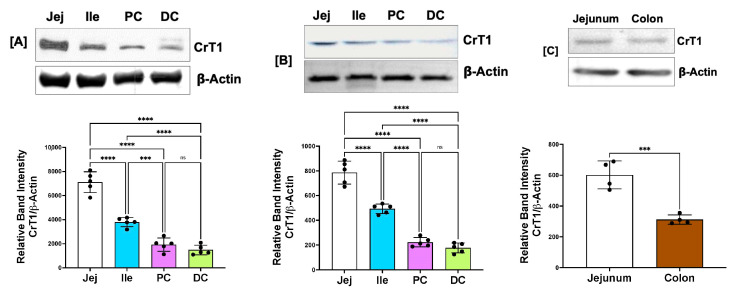
Protein levels of CrT1 along the length of mouse (**A**), rat (**B**), and human (**C**) intestinal mucosa. Tissue lysates prepared from scrapped mucosa were subjected to 10% SDS-polyacrylamide gel, transferred to PVDF membrane, and probed with anti-CrT1 antibody. The upper panels are representative blots showing the relative expression of CrT1 in different regions of the intestine with β-actin as the loading control. For mice samples, 75 µg protein was loaded per well of the gel, and for rats and human samples, 50 µg was loaded. Lower panels show the results of densitometric analysis of band intensities of CrT1 normalized to that of β-actin. Data represent mean ± SEM. *** *p* < 0.001, **** *p* < 0.0001 and ns = not significant between groups as indicated.

**Figure 3 ijms-25-06537-f003:**
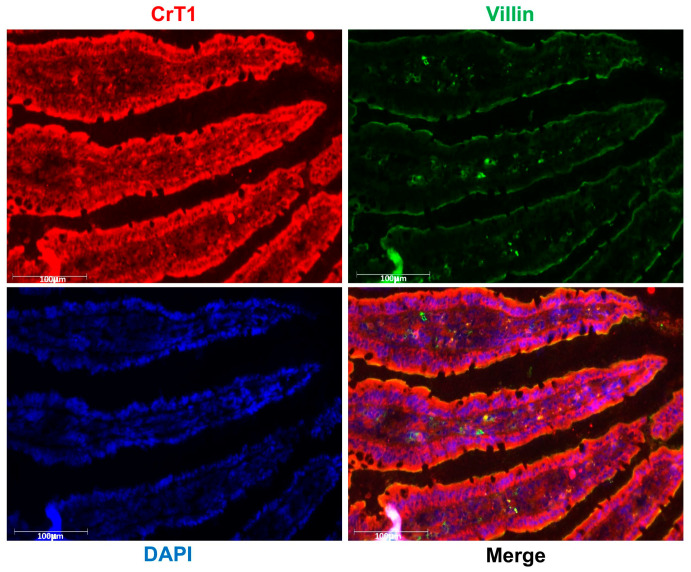
CrT1 is localized to the luminal (brush border) membrane in rat jejunum. OCT sections of jejunum were immunostained for CrT1 (red), the apical membrane marker villin (green), and nuclei (blue) as described in Methods. Uniform expression of CrT1 in cells along the length of the villi is shown in red. BBM localization of villin in cells along the length of the villi is shown in green. Colocalization of CrT1 with villin in the BBM of villus cells is shown in the merged image. A representative image is shown. Scale bar—100 µm.

**Figure 4 ijms-25-06537-f004:**
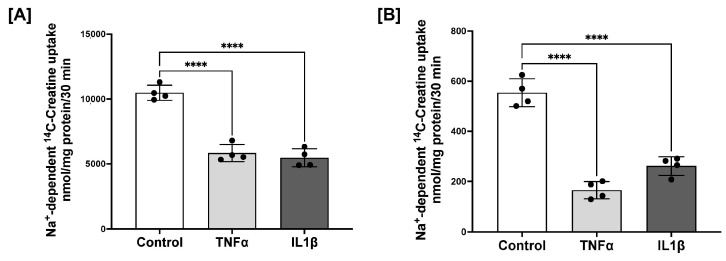
Proinflammatory cytokines inhibit CrT1 function (Na^+^-dependent ^14^C-creatine uptake) in Caco-2 (**A**) and IEC-6 (**B**) cells. Post-confluent cell monolayers grown on 12-well transwells were treated with TNFα or IL1β (10 ng/mL) from the basolateral compartment for 24 h. Na^+^-dependent uptake of ^14^C-creatine was determined in the presence or absence of the competitive CrT1 substrate β-guanidinopropionic acid (GPA, 1 mM). Results are expressed as nmoles-mg protein^−1^ × 30 min^−1^. Data represent mean ± SEM. **** *p* < 0.0001 between groups as indicated.

**Figure 5 ijms-25-06537-f005:**
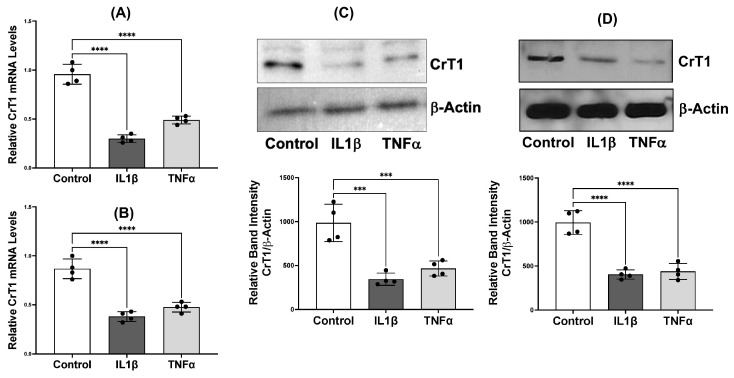
Proinflammatory cytokines decrease mRNA and protein levels of CrT1 in IECs. Caco-2 and IEC-6 cells grown on transwell inserts were treated with TNFα or IL1β (10 ng/mL) from the basolateral compartments for 24 h. mRNA levels of CrT1 in total RNA from Caco-2 (**A**) and IEC-6 (**B**) cells were amplified by real-time QRT-PCR using species-specific (human or rat) primers for the *CrT1* gene. Similarly, protein levels of CrT1 in the total lysates of Caco-2 ((**C**), **upper panel**) and IEC-6 ((**D**), **upper panel**) were determined by Western blot. Lower panels show the respective densitometric analysis of the band intensities of CrT1 normalized to that of β-actin. Data represent mean ± SEM. *** *p* < 0.001 and **** *p* < 0.0001 between groups as indicated.

**Figure 6 ijms-25-06537-f006:**
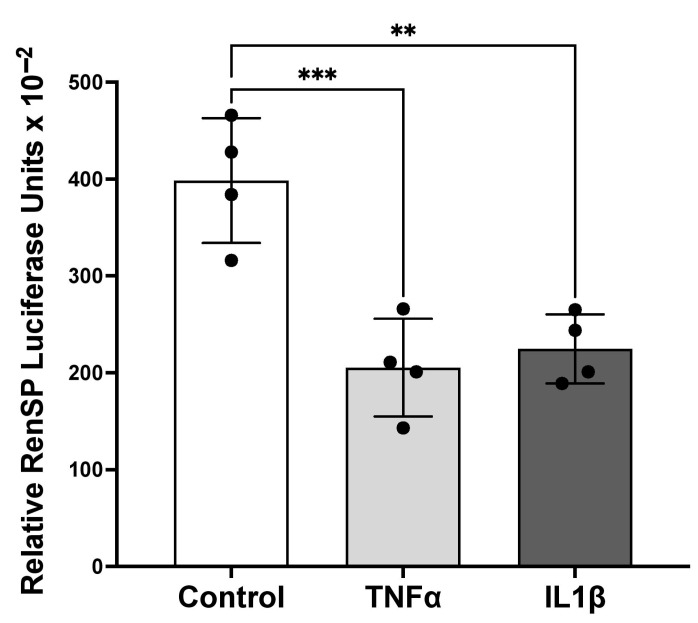
TNFα and IL1β treatments decrease CrT1 promoter activity in Caco-2 cells. A 1029 bp fragment of the human CrT1 promoter region was transfected into Caco-2 cells using Lipofectamine 3000 (Invitrogen, Waltham, MA, USA). Twenty-four hours post-transfection, cells were treated with TNFα or IL1β (10 ng/mL) for 24 h. Promoter activity (luciferase units) was determined as described in the Methods section. Data represent mean ± SEM. ** *p* < 0.01 and *** *p* < 0.001 between groups as indicated.

**Figure 7 ijms-25-06537-f007:**
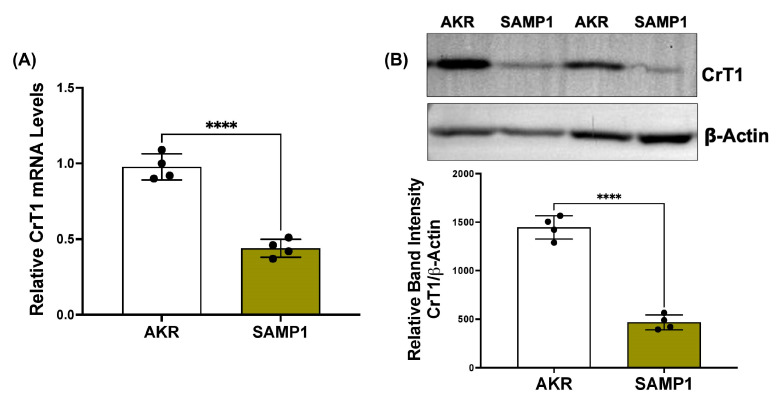
CrT1 expression is decreased in vivo in SAMP1 small intestinal mucosa compared to AKR mice. (**A**) CrT1 mRNA levels in total RNA extracted from SAMP1/AKR small intestinal mucosa were measured by QRT-PCR, utilizing gene-specific mouse primers. Data represent the relative expression of CrT1 normalized to respective GAPDH mRNA (internal control). (**B**) CrT1 protein levels in total lysate extracted from SAMP1/AKR small intestinal mucosa were measured by Western blot. The upper panel shows a representative blot. The lower panel shows the densitometric analysis of the band intensities of CrT1 normalized to that of β-actin. Data represent mean ± SEM. **** *p* < 0.0001 between groups as indicated.

**Figure 8 ijms-25-06537-f008:**
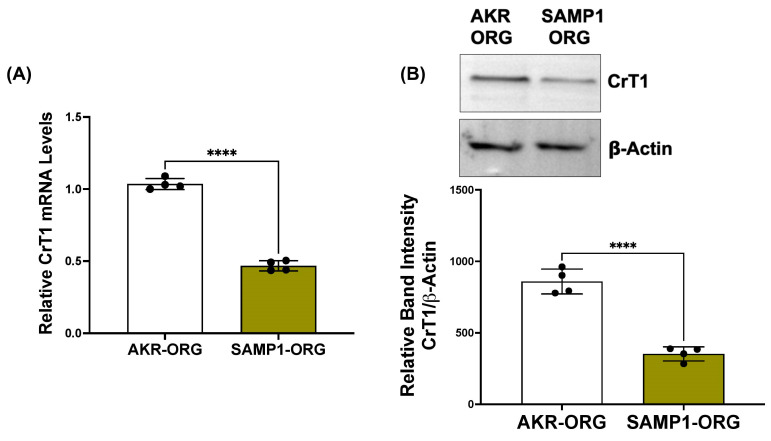
CrT1 expression is decreased in SAMP1 small intestinal organoids compared to AKR organoids. (**A**) CrT1 mRNA levels in total RNA extracted from SAMP1/AKR small intestinal organoids were measured by QRT-PCR, utilizing gene-specific mouse primers. Data represent the relative expression of CrT1 mRNA normalized to respective β2-microglobulin (B2M) mRNA (internal control). (**B**) CrT1 protein levels in total lysates made from SAMP1/AKR small intestinal organoids were measured by Western blot. The upper panel shows a representative blot. The lower panel shows the densitometric analysis of the band intensities of CrT1 normalized to that of β-actin. Data represent mean ± SEM. **** *p* < 0.0001 between groups as indicated.

**Figure 9 ijms-25-06537-f009:**
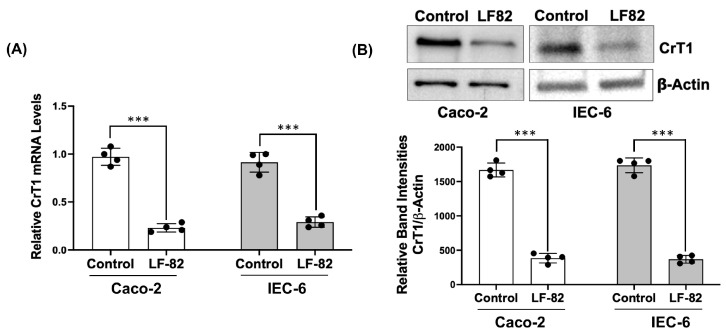
AIEC infection decreases CrT1 expression in vitro. Cell monolayers (Caco-2 or IEC-6) were treated from the apical surface for 8 h with a 1:10 diluted bacterial suspension of an exponentially grown overnight culture of AIEC LF-82 showing an absorbance_(600 nm)_ = 0.2. (**A**) CrT1 mRNA levels in total RNA extracted from Caco-2/IEC-6 cells were measured by QRT-PCR, utilizing gene-specific human/rat primers. Data represent the relative expression of CrT1 mRNA normalized to respective GAPDH mRNA (internal control). (**B**) CrT1 protein levels in total lysates of Caco-2/IEC-6 cells were measured by Western blot. The upper panel shows a representative blot. The lower panel shows the densitometric analysis of the band intensities of CrT1 normalized to that of β-actin. Data represent mean ± SEM. *** *p* < 0.001 between groups as indicated.

**Figure 10 ijms-25-06537-f010:**
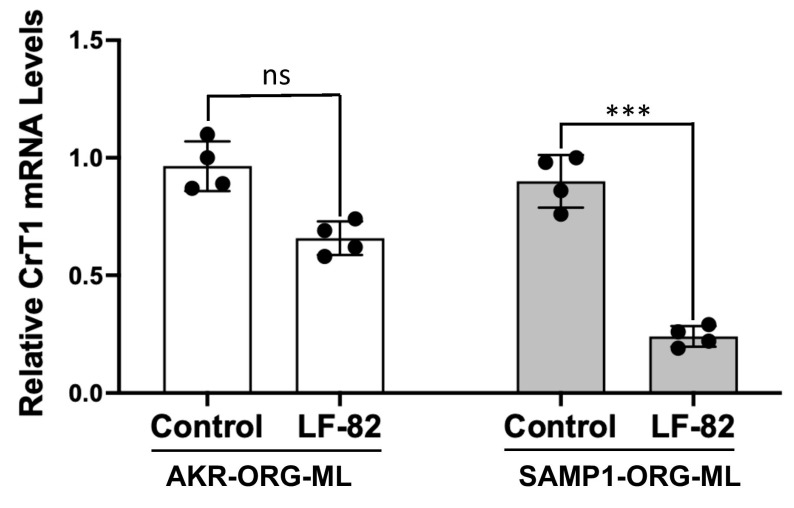
AIEC infection diminishes CrT1 in SAMP1 organoid-derived monolayers with no significant effects on AKR organoid monolayers. Two-dimensional monolayers of crypt-derived small intestinal organoids generated from SAMP1/AKR mice were treated from the apical surface for 8 h with a 1:10 diluted bacterial suspension of an exponentially grown overnight culture of AIEC LF-82 showing an absorbance_(600 nm)_ = 0.2. (A) CrT1 mRNA levels in total RNA extracted from SAMP1/AKR organoid monolayers (ORG-ML) were measured by QRT-PCR, utilizing gene-specific mouse primers. Data represent the relative expression of CrT1 mRNA normalized to respective β2-microglobulin (B2M) mRNA (internal control). Data represent mean ± SEM. *** *p* < 0.001 and ns = not significant between groups as indicated.

**Figure 11 ijms-25-06537-f011:**
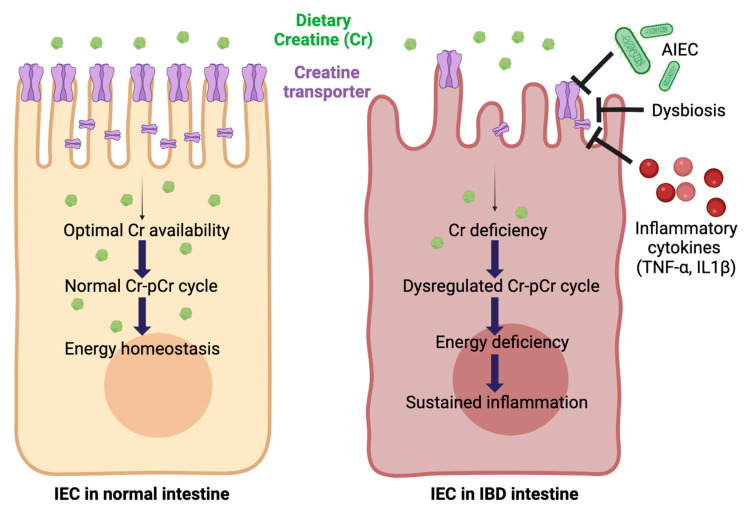
Schematic representation of the potential cause-and-effect relationship between CrT1 deficiency and IBD-associated chronic mucosal inflammation. CrT1 plays a critical role in maintaining energy homeostasis in intestinal epithelial cells (IECs) via mediating the absorption of dietary creatine (Cr), a potent energy nutrient. Optimal Cr availability ensures proper functioning of the creatine–phosphocreatine (Cr-pCr) shuttle that supplies energy to IECs. In inflammatory bowel disease (IBD), elevated levels of proinflammatory cytokines, such as TNFα and IL1β, diminish CrT1 expression and function. Further, IBD-associated dysbiosis triggers colonization of the inflamed mucosa by several opportunistic pathogens, adherent invasive *E. coli* (AIEC) being a predominant one. AIEC infection further downregulates CrT1 which results in Cr deficiency and impaired Cr-pCr shuttle. Thus, IECs become energy deficient at a time of high energy demand to combat the effects of active inflammation and dysbiosis. Therefore, impaired Cr transport into IECs could be a major contributing factor in sustaining chronic inflammation in IBD as well as for its relapsing and recurring episodic nature.

**Table 1 ijms-25-06537-t001:** Gene-specific primers used for real-time PCR analysis of mRNA levels (F: forward primer; R: reverse primer).

Gene	Primer Sequence (5′-3′)
Human *slc6a8*	F: CTTCATCATGTCGTGCGTGG R: GCGATCAGGACGTAGGGAAT
Mouse *slc6a8*	F: TGGGGGTAAGGGTGGAATGTA R: TGTCATCCATGAAGCGGTCA
Rat *slc6a8*	F: CCTCAACTGGGAAGTGACCC R: AGGACCACGTAGGGGAATGT
Human *GAPDH*	F: TGCACCACCAACTGCTTAGC R: GGCATGGACTGTGGTCATGAG
Mouse *GAPDH*	F: TGTGTCCGTCGTGGATCTGA R: CCTGCTTCACCACCTCTTGAT
RAT *GAPDH*	F: GGCAAGTTCAACGGCACAG R: CGCCAGTAGACTCCAC GAC
Mouse *β2-Microglobulin*	F: CTGGTGCTTGTCTCACTGAC R: GTTCAGTATGTTCGGCTTCC

## Data Availability

The authors confirm that the data supporting the findings of this study are available within the article.

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
