# Peer review of "Intestinal Epithelial Creatine Transporter SLC6A8 Dysregulation in Inflammation and in Response to Adherent Invasive E. coli Infection"

_ijms, 2024, doi:10.3390/ijms25126537_

Round 1

Reviewer 1 Report

Comments and Suggestions for Authors

The study of the expression/immunoekspression of the gene encoding/protein the creatinine transporter (CrT1) in the gastrointestinal tract of mouse, rat, and human was the subject of a paper by Sawant et al. The work is clinically relevant and performed using standard molecular biology tools (WB, IHC, rtPCR). Unfortunately, the paper contains an embarrassingly poorly described methodological side of the study. In addition, the paper unfortunately contains deficits related to anatomical terminology, which largely prevent it from being published as is.

1 Anatomically, the intestines are two - small intestine and large intestine - while there is no single common term for the two intestines. The authors, using the term “intestine” in the title and text of the paper, do not specify exactly which part of the digestive tract they mean.  The small intestine is divided into the duodenum, jejunum and jejunum. Why did the authors not include the duodenum in their study?

2 The large intestine is divided into the cecum, colon and rectum while the proximal and distal colon are not divided. Such a division is not acceptable.

3. Figures 2A-C, 5C-D, 7B, 8B, 9B - the supplementary materials lack full uncropped WB gels with the ladder visible. Those included in the mentioned figures are insufficient.

4. Figure 3 - immunofluorescence images are blurry and uninformative. For the less familiar reader, it is not clear what he is actually looking at. Explanatory arrows are missing. Scale bars are illegible.

5. gene names should be written in italics

6. line 378 - there is no data on the antibodies used (codes, species). In addition, the authors in no way carried out the control procedure of antibodies (preadsorption tests) therefore did not prove their specificity.

7. line 390 - the description of the statistical analysis is too enigmatic. Was a normal distribution tested and how? What was done when a normal distribution was not found?

Reviewer 2 Report

Comments and Suggestions for Authors

In this study, the authors tried to investigate the differential expression of creatine transporter (CrT1) along the length of mouse, rat, and human intestine and determine significant downregulation of CrT1 expression and function in inflammatory bowel disease (IBD)-associated inflammation and in response to AIEC infection via distinct mechanisms. The authors concluded that creatine (Cr) deficiency in intestinal epithelial cells (IECs) secondary to CrT1 dysregulation could be a key factor contributing to IBD pathogenesis.

Comments:

This is an interesting study. The reviewer has some concerns as follows:

1.     In the Introduction, the authors described that they report for the first time the differential expression of CrT1 along the length of mouse, rat, and human intestine. However, in the Methods, there are no information for IACUC for animal experiments and IRB for human samples. Please provide the related information.

2.     In the Methods - 5.7. Western Blotting, the category number and dilution ratio for primary antibodies can be described.

3.     In Figure 2, the band intensities of protein levels of CrT1 along the length of mouse, rat, and human intestinal mucosa are very different. Is there any species difference in this situation? This issue can be discussed.

4.     In Figure 3, the fonts for scale bars are too small to be legible. Please increase the font size or explain it in the figure legend.

5.     In Figure 5, the concentrations for TNFα or IL1β treatment can be described in the figure legend.

6.     In Figure 6, the title “TNFα IL1β treatments…” can be changed to “TNFα and IL1β treatments…”. Moreover, the concentrations for TNFα or IL1β treatment can be described in the figure legend.

7.     In the legend of Figure 9, the description for “Cell monolayers (Caco-2 or IEC-6) were treated with OD600 nm = 0.02 of overnight grown LF-82 strain of AIEC for 8 h” seems unclear. What does “OD600 nm = 0.02” mean?

8.     Overall, this manuscript needs a revision before it can be accepted.

Reviewer 3 Report

Comments and Suggestions for Authors

This original research article manuscript is an interesting study on SLC6A8 regulation in intestinal epithelial cells in healthy and non-healthy models. It seems to be overall novel, and adds important knowledge to the field, while also being well discussed and structured. Thus, I only advise on the following alterations:

- The introduction section is not well supported by the appropriate references, some whole paragraphs do not contain 1 single reference, this should be corrected;

- The number of references to support what is being said is quite short, this should be improved, for added robustness;

- The graphical/visual aspect of the article should be improved, namely by giving bar graphs different colors, which will also help the reader distinguish between different data;

- In vitro-in vivo data correlation should be further explored and discussed;

- The authors should discuss Cr deficiency in IECs secondary to CrT1 dysregulation not only in what concerns IBD pathogenesis, but also as a possible therapeutic target (it is mentioned in passing but should be better discussed, deepened);

- In the introduction section, more should be said on IBD pathogenesis (especially molecular aspects) and management (current therapies, and their limitations);

- More should be said on complementary ways of managing IBD, namely the simultaneous use of pharmacological and nutritional methods;

- In the introduction section, a representative image should be produced and added regarding IBD pathogenesis, the role of Cr, and Adherent Invasive E. coli as a relevant opportunistic pathogen, for summarization and better reader understanding;

- Figure 3 should be provided in better quality (resolution), and should also be better explained both in the text and in the respective figure caption.

Round 2

Reviewer 1 Report

Comments and Suggestions for Authors

Scientific work must take into account anatomical nomenclature. I am not interested in how the authors divide the large intestine for themselves. Anyone potentially reading such a paper must have clear data and not wonder what the authors are actually writing about. My other comments were treated superficially or not addressed at all. Further there are no antibody specificity tests at least.
My previous opinion remains the same.

Reviewer 2 Report

Comments and Suggestions for Authors

This revised manuscript has a great improvement. No further comments.